# Structurally Preserved *Liquidambar* Infructescences, Associated Pollen, and Leaves from the Late Oligocene of the Nanning Basin, South China

**DOI:** 10.3390/plants13020275

**Published:** 2024-01-17

**Authors:** Sheng-Lan Xu, Natalia Maslova, Tatiana Kodrul, Nikita Zdravchev, Vasilisa Kachkina, Xiao-Yan Liu, Xin-Kai Wu, Jian-Hua Jin

**Affiliations:** 1School of Life Sciences/School of Ecology, Sun Yat-sen University, Guangzhou 510275, China; xushlan3@mail.sysu.edu.cn; 2Borissiak Paleontological Institute, Russian Academy of Sciences, Moscow 117997, Russia; paleobotany_ns@yahoo.com; 3Geological Institute, Russian Academy of Sciences, Moscow 119017, Russia; kodrul@mail.ru; 4Tsitsin Main Botanical Garden, Russian Academy of Sciences, Moscow 127276, Russia; zdravchevnikita@yandex.ru; 5Department of Higher Plants, Faculty of Biology, M.V. Lomonosov Moscow State University, Moscow 119991, Russia; kachkina.v@gmail.com; 6School of Geography, South China Normal University, Guangzhou 510631, China; lxy_0628@163.com; 7State Key Laboratory of Palaeobiology and Stratigraphy, Nanjing Institute of Geology and Palaeontology, Nanjing 210008, China

**Keywords:** *Altingiaceae*, reproductive structures, pollen grains, micro-computed tomography, *Oligocene*, China

## Abstract

*Liquidambar* L. is a significant constituent of the Cenozoic flora in the Northern Hemisphere. Currently, this genus exhibits a discontinuous distribution across Asia and North America, with the center of diversity being in southeastern Asia. This study presents the first occurrence of *Liquidambar* in the Oligocene of South China. Fossil sweetgum infructescences, associated pollen, and leaves have been found in the Nanning Basin, Guangxi. A new species, *Liquidambar nanningensis* sp. nov., is described based on the morphological and anatomical characteristics of three-dimensionally preserved infructescences. The *Liquidambar* fossils from the Nanning Basin show a combination of features indicative of the former genera of Altingiaceae, *Altingia, Liquidambar* s. str., and *Semiliquidambar*. The new occurrence expands the taxonomic and morphological diversity of the Paleogene *Liquidambar* species in South China.

## 1. Introduction

Following a taxonomic revision of Altingiaceae provided by Ickert-Bond and Wen [1], the genera *Liquidambar*, *Altingia* Noronha, and *Semiliquidambar* H.-T.Chang were combined into a single genus *Liquidambar* with 15 extant species. In addition to the molecular evidence which suggests that *Altingia* and *Semiliquidambar* are nested within *Liquidambar*, the three former genera of Altingiaceae share several common morphological characteristics: capitate reproductive structures, woody infructescences, bicarpellate fruits with a syncarpous ovary, pantoporate pollen grains, semicraspedodromous venation of secondary veins, uniformly serrated leaf margins, and tooth morphology [1,2,3]. The main differences between these former genera are related to their deciduous or evergreen strategies and wind pollination. *Liquidambar* species are deciduous wind-pollinated trees with palmately 3–7 (or more)-lobed, rarely entire leaves. Species of *Altingia* are represented by evergreen trees with entire leaves (lanceolate to ovate or obovate). *Semiliquidambar*, previously considered as an intergeneric hybrid between *Liquidambar* and *Altingia*, combined three species of deciduous or evergreen trees with entire, asymmetrically lobed or three-lobed leaves. Morphological characteristics that distinguish these former genera may represent adaptation to different habitats, temperate sites for *Liquidambar,* and tropical to subtropical regions for *Altingia* and *Semiliquidambar* [1,2,3].

The evolutionary history of the family Altingiaceae has become increasingly clear in recent years due to the recovery of new fossil records. Most of the *Liquidambar* fossils are lobate leaf fossils widely represented in the Paleogene and Neogene of North America, Asia, and Europe [4,5,6,7,8,9,10,11,12,13,14,15,16,17,18,19,20,21,22,23,24,25,26,27,28,29,30,31,32,33,34]. The oldest *Liquidambar* pollen [35,36,37,38], wood [39,40], and leaf morphotypes [30] are known from the Paleocene. However, pantoporate *Liquidambar*-like pollen has been reported since the Campanian–Maastrichtian [41]. The earliest leaf fossils assigned to *Liquidambar* are described from the Late Cretaceous of North America [6], but the altingioid affinity of these fossils has been questioned [14,42]. 

Reproductive structures comparable to those of *Liquidambar*, but differing in some microstructural features of the capitate inflorescences/infructescences and pollen morphology, are known from the upper Turonian (*Microaltingia* Zhou, Crepet et Nixon; *Paleoaltingia* Y.J. Lai, Gandolfo, and Crepet et Nixon) [43,44] and lower Coniacian (*Protoaltingia* Scharfstein and Stockey et Rothwell) [45] of North America. Capitate infructescences of the Eocene genus *Steinhauera* C. Presl related to Altingiaceae are known from some localities of Europe [46,47,48]. Three-dimensionally preserved reliable infructescences of *Liquidambar* have been described only from the Miocene [33,42]. Numerous capitate infructescences assigned to *Liquidambar* are preserved as compressions in the Cenozoic of Asia, Europe, and North America [8,10,18,20,22,30,32,49]. Furthermore, co-occurrences of *Liquidambar* leaves and reproductive structures are of considerable interest [18,22,30,32,33,50]. In China, altingioid megafossils are known from the Eocene [30,31,32,51,52,53], Miocene [7,28,33,50,54], and Upper Pleistocene [54]. Until now, megafossils of *Liquidambar* from the Oligocene of China have not been reported. Here, we describe structurally preserved infructescences, associated pollen, and leaves from the upper Oligocene of the Nanning Basin, South China. The new occurrence of fossil *Liquidambar* expands our understanding of the morphological and taxonomic diversity of this genus in the geological past. Additionally, due to the current reassessment of the Altingiaceae taxonomy, the revealing of possible fossil evidence is of great importance.

## 2. Results

### 2.1. Systematic Botany

Family: Altingiaceae Horan., 1841

Genus: *Liquidambar* L., 1753

Species: *Liquidambar nanningensis* Xu, Zdravchev, N. Maslova et Jin, sp. nov. 

Holotype: NNEZ-15a, NNEZ-15b, designated here (Figure 1A–D), a head part and counterpart. 

Additional specimens: NNEZ-17a, NNEZ-21a, NNEZ-21b (Figure 2A–F), NNEZ-28 (Figure 3A), NNEZ-32 (Figure 3B), NNEZ-35 (Figure 3C), NNEZ-42 (Figure 3D), NNEZ-43, and NNEZ-48

Etymology: The specific epithet is derived from the Nanning Basin.

Locality: Santang Town, Nanning, Guangxi

Stratigraphic position: Yongning Formation, Nanning Basin.

Geological age: Late Oligocene.

Repository: The Museum of Biology, Sun Yat-sen University, Guangzhou, China.

Diagnosis: Capitate infructescence from 15 to 24 mm in diameter comprising 10–12 fruits. The fruits are bilocular and the carpels are fused basally and free distally. Style absent. Perianth lacking. Epidermal cells of carpel wall with oblique end walls.

Description. The capitate infructescences are somewhat flattened as a result of fossilization. The heads are rounded in their outline, ranging from 15 to 24 mm in diameter (Figure 1A,B, Figure 2A and Figure 3A–D). Fruit numbers per head, according to the fruit scars, apparently do not exceed 10–12. The maximum diameter of individual fruits is 6–7 mm, and the length is 10.5–11.5 mm. The fruits are helically arranged, bilocular with two luniform carpels, from broadly obovoid to almost obconical in outline, free distally, and fused basally at a distance equal to half the length of the carpel (Figure 1A–D, Figure 2A–F and Figure 3A–D). The carpel wall epidermal cells possess oblique end walls (Figure 4A). Fruits adhere tightly to each other, and distinct peripheral rims of tissues between adjacent fruits are not observed. Style remnants are not revealed. Perianth is lacking.

Remarks. Pollen grains of *Liquidambar* were observed on the surface of *Liquidambar nanningensis* carpels (Figure 4B,C). Additionally, dispersed pollen grains of the same type were found in association with the infructescences (Figure 4D–F and Figure 5A–F). The fruiting body of an endophytic micromycete (Figure 6D) and fungal spores (Figure 6A–C) occur on the *L. nanningensis* carpel surface.

### 2.2. Liquidambar sp. 1

The pollen grains are pantoporate (38.2 × 25.8 µm to 30.1 × 23.2 µm) and circular to elliptical in outline. The pores are circular in shape to slightly elliptical, with smooth margins. The exine is foveolate; perforations in the reticulum of the tectum vary in size and shape, from rounded to somewhat elongated or irregular. The exine bears numerous small verrucae of irregular size and shape on the surface of the tectum and pore membranes (Figure 4D–F and Figure 5A–F).

Material. NNEZ-42

### 2.3. Liquidambar sp. 2

Two fragmentary preserved leaves of *Liquidambar* are associated with infructescences. The leaves are simple and petiolate, with a cuneate base (Figure 7A–C). Their lamina length reaches approximately 7–8 cm. The leaf margin is serrate. Teeth are spaced at some distance from the leaf base (Figure 7A–C). The teeth are appressed and concave/retroflexed, with a more prominent proximal flank and rounded sinuses (Figure 7D). Tooth apices are non-specific. Venation is suprabasal actinodromous and semicraspedodromous, with three primaries (Figure 7A–D). The midvein is straight. Lateral primary veins are equal in thickness to the midvein and curved upward.

Material. NNEZ-45a-1, NNEZ-45a-2, and NNEZ-45b.

## 3. Discussion

### 3.1. Morphological Comparison of Studied Fossils with Extant and Fossil Asian Liquidambar Species

#### 3.1.1. Reproductive Structures

Head. The majority of fossil *Liquidambar* infructescences are preserved as compressions. Although the reproductive structures of extant representatives of families Altingiaceae and Platanaceae differ significantly, fossil altingioid and platanoid compressed fruiting heads are often very similar in gross morphology [44,55], especially those with persistent styles [10,20,50,56]. In this case, the associated leaves and pollen provide indirect evidence that the fossil inflorescences/infructescences may belong to the same taxon. The presence of visible rims of tissues between adjacent fruits seems to be a distinctive feature of altingioid capitate reproductive organs with absent or inconspicuous styles [30,32]. 

Woody infructescences (heads) of extant species previously considered within the former genera *Altingia* and *Liquidambar* s. str. differ from one another in terms of the number of fruits per head, the presence or absence of a cuplike bract at the base of the head, the thickness and ornamentation of the peripheral rim around individual fruits, the degree of development, the length and shape of styles, the types of extrafloral phyllomes, variation in the anatomy of the fruit walls, and seed morphology [2,57].

Fruiting heads from the Nanning Basin can be accurately assigned to *Liquidambar* based on the following morphological characteristics: the rounded shape of the capitate infructescences composed of few bilocular fruits fused basally and free distally and the lack of styles and perianths, as well as carpel walls consisting of tetragonal cells with oblique end walls. In the number of fruits per head and the absence of styles, the infructescences of *L. nanningensis* are similar to those of the Asian extant species of *Liquidambar* previously considered within the genus *Altingia*, e.g., *L. gracilipes* (Hemsl.) Ickert-Bond et J. Wen (previously *A. gracilipes* Hemsl.), *L. siamensis* (Craib) Ickert-Bond et J. Wen (previously *A. siamensis* Craib), and *L. yunnanensis* (Rehder et E.H. Wilson) Ickert-Bond et J. Wen (previously *A. yunnanensis* Rehder et E.H. Wilson). The main morphological difference between their infructescences is the absence of bracts at the base of *L. nanningensis* heads. All other extant *Liquidambar* species differ in their large number (up to 40) of fruits per head. 

The peripheral rim between adjacent fruits in *L. nanningensis* is not prominent, and fruits are tightly pressed against one another. With these features, *L. nanningensis* is more similar to the extant species *L. styraciflua* L. and *L. acalycina* H.-T. Chang. However, the new fossil species differ in fruit number per head and the absence of prominent styles. The extant species previously considered within the genus *Liquidambar* s. str. [2] have carpel wall cells with oblique end walls, a feature shared with *L. nanningensis*. 

Among fossil *Liquidambar* infructescences, permineralized female reproductive structures from the Middle Miocene of Yakima Canyon, Washington state, USA, are the most well studied [42]. In China, the only anatomically studied infructescences are three-dimensionally preserved fruiting heads of *L. fujianensis* J.L. Dong et B.N. Sun from the Middle Miocene Fotan Group, southeastern China [33]. Infructescences of *L. fujianensis,* 6 to 25 mm in diameter, are composed of more than 20 individual bilocular fruits with persistent styles. The carpel epidermal cells in *L. fujianensis* show oblique end walls, similar to those in fossil *L. nanningensis* and *L. ovoidea* Kachkina, N. Maslova, Kodrul et Jin from the Eocene of Hainan Island [32], as well as in the extant *Liquidambar* species. The new species *L. nanningensis* differs from *L. fujianensis* in having a smaller number of fruits per head and lacking styles. 

The capitate infructescences associated with leaves of *L. maomingensis* N. Maslova, Kodrul, Song et Jin from the Eocene Huangniuling Formation, Maoming Basin, South China [30], are preserved as impressions/compressions. These infructescences are rounded and from 13 to 16 mm in diameter. The fruit number per head does not exceed 10, and they are bilocular and separated distinctly from each other by slightly raised, smooth tissue. A perianth is lacking and the styles are tiny. The new species *L. nanningensis* differs from *L. maomingensis* in having a somewhat larger number of fruits per head, lacking both styles and a peripheral rim of tissue surrounding individual fruits.

The capitate infructescences of *L. ovoidea* associated with leaves of *L. hainanensis* Kachkina, N. Maslova, Kodrul, et Jin from the middle–upper Eocene Changchang Formation, Hainan, South China [32], are ovoid to subglobose and from 13 to 18 mm in diameter. The fruit number per head does not exceed 16. Bilocular fruits lack perianth and styles, and they are separated from each other by the slightly raised tissue. The new species *L. nanningensis* differs from *L. ovoidea* in the globose infructescences and the absence of distinct rims of tissue between adjacent fruits.

In contrast to *L. nanningensis, L. miosinica* Hu et Chang reported from the Middle Miocene Shanwang flora, Shandong Province, China [58], and the Late Miocene Xiananshan flora, Zhejiang Province, China [50] are characterized by infructescences with prominent styles. Fruiting heads associated with the leaves of *L. protopalmata* (K. Suzuki) Uemura from the Neogene of northeast Honshu, Japan, also possess carpels with long persistent styles [20]. An incompletely preserved small head of *Liquidambar* sp. associated with leaves of *L. miosinica* was reported from the Lower Miocene Nakamura Formation, the southern part of Gifu Prefecture, Japan [59].

Pollen. Data on the pollen associated with fossil *Liquidambar* infructescences are very scarce. In Asia, such associations are known from the Oligocene and Miocene of Kazakhstan [36,60]. In China, the Eocene leaves of *L. maomingensis* from the Maoming Basin [30] and *L. hainanensis* from the Changchang Basin [32] co-occur with *Liquidambar*-type pollen obtained from palynomorph assemblages. Pollen grains from the Changchang Basin [32] are very similar to those of extant *L. orientalis,* consistent with the similarity of *L. hainanensis* and *L. orientalis* leaves. Pollen grains of *Liquidambarpollenites* from the Maoming Basin have only been studied using LM [61], which limits their detailed comparison with those of fossil and extant species. 

The palynomorph assemblage from the Yongning Formation of the Nanning Basin contains *Liquidambarpollenites* pollen [62,63]. Pollen grains of *Liquidambarpollenites* have also been reported from the Oligocene palynomorph assemblage of the Lühe Basin, Yunnan Province, southwestern China [64]; however, no other Altingiaceae fossils were found there.

Dispersed pollen grains of *Liquidambar* from the palynomorph assemblage of the Yongning Formation (Figure 4D–F and Figure 5A–F) and pollen grains attached to the *L. nanningensis* carpels (Figure 4B,C) are identical at least in their size, shape, pore outlines, and foveolate exine. These pollen grains were most probably produced by the same plant. Despite the fact that the morphology of *Liquidambar* pollen grains appears uniform, different species have some distinctive features. The size of pollen grains, the number of pores, their shape, diameter, and margin peculiarities vary somewhat between species [65]. Pollen grains from the Nanning Basin are comparable to those of the extant species *Liquidambar gracilipes* (previously *Altingia gracilipes*). Similarities are based on the frequency and size of verrucae on the tectum surface, perforation size and shape (shape varies from rounded to somewhat elongated or irregular), distinct smooth pore margins with rare minute verrucae, and the disintegration of the tectum with relatively large verrucate granules on the pore membrane. However, pollen of the former genus *Altingia* are characterized by round pores, while fossil pollen grains from the Nanning Basin have both round and slightly elliptic pores. Round and occasionally slightly elliptic pores with smooth margins are common for *Liquidambar formosana* [36,66].

#### 3.1.2. Leaves

Despite incomplete preservation, some fossil leaves from the Nanning Basin exhibit a number of specific morphological features characteristic of the leaves of *Liquidambar*: a long thin petiole, distinctive appressed, concave/retroflexed marginal teeth, and suprabasal actinodromous semicraspedodromous venation. These are the only *Liquidambar* leaves known from the Oligocene of China. Previously, fragmentary trilobate leaves with basal actinodromous venation and regularly spaced marginal teeth assigned to *Liquidambar* sp. were reported from the Eocene of China [51,52]. Recently, three *Liquidambar* species, *L. maomingensis* [30], *L. bella* N. Maslova et Kodrul [31], and *L. hainanensis* [32], were described from the Eocene of South China. Leaves of *L. maomingensis* show high morphological variability, ranging from three-lobed to unlobed leaves with transitional morphotypes having one small lobe. While leaves of *L. maomingensis* have exclusively basal venation, two other Chinese Eocene species, *L. bella* and *L. hainanensis*, are distinguished by a combination of morphotypes with both basal and suprabasal venation.

Suprabasal venation is known in the extant species *Liquidambar chingii* (Metcalf) Ickert-Bond et J. Wen (previously *Semiliquidambar cathayensis* H.T. Chang) and *L. caudata* (H.T. Chang) Ickert-Bond et J. Wen (previously *Semiliquidambar caudata* H. T. Chang) (Figure 8). All other extant and fossil species of *Liquidambar* with lobed laminae have lateral primary veins with a basal origin. Similar to leaves of *L. bella* and *L. hainanensis,* leaves of *Liquidambar* sp. associated with heads of *L. nanningensis* are characterized by suprabasal venation.

Leaves of *Liquidambar* sp. from the Nanning Basin are most similar to those of *L. hainanensis* based on the irregular distribution of teeth along the margin and, in particular, a partly untoothed leaf base. The extant species *L. caudata* is also characterized by an untoothed lower part of the lamina (Figure 8). The leaves of *L. fujianensis* from the Middle Miocene of southeastern China [33] are predominantly three-lobed, although five-lobed and unlobed leaves also occur. The lateral primary veins in lobed leaves of *L. fujianensis* always diverge from the midvein basally. 

### 3.2. Fungi Associated with L. nanningensis Heads

A fungal fruiting body was found on the carpel surface of *L. nanningensis* (Figure 6D). The body is oval and 52 × 37 µm in size, with a central, weakly expressed, rounded ostiole 10 µm in diameter. The ostiole zone is cracked. Chains of spores were observed on the carpel outer cuticle near the fruiting body (Figure 6A–C). Possibly, these chains represent the remains of asci. Spores are aseptate, round, or slightly oval, ranging in size from 13 × 9 to 18 × 11 µm.

Many micromycetes of the Ascomycota live on lignified parts of plants, with them being saprophages [67]. Two species of ascomycetes were described from the heads of extant *Liquidambar*. *Xylaria persicaria* (Schwein. Fr.) Berk. et M.A. Curtis (Xylariaceae) differs in having aseptate ellipsoid ascospores with a long spiraling germ slit [68], while *Massaria inquinans* (Tode) De Not. (Massariaceae) is characterized by cylindrical or fusoid uni-, bi-, or triseriate ascospores [69]. The poor preservation of the single ascoma from the infructescence of *L. nanningensis* prevents its correct taxonomic classification.

### 3.3. Taxonomic Significance of New Liquidambar Fossils from the Nanning Basin

Recently, the occurrence of a group of polymorphic *Liquidambar* species was assumed in the Eocene of South China, including Hainan Island [30,31,32]. Leaves of three middle-upper Eocene species, *L. maomingensis* and *L. bella* from the Maoming Basin of South China [30,31] and *L. hainanensis* from the Changchang Basin, Hainan [32], possess the mosaic of morphological features characteristic of several extant species of *Liquidambar* (including species of the former genera *Altingia* and *Semiliquidambar*). Capitate infructescences associated with *L. maomingensis* leaves are similar to those of the former *Altingia* species. Similarly, polymorphic leaves of *L. hainanensis* combine the leaf morphological features of two genera, *Liquidambar* s. str. and former *Semiliquidambar*. These fossil leaves co-occur with dispersed pollen grains of the *Liquidambar* type and capitate infructescences of *L. ovoidea* which are similar to the infructescences of extant *Liquidambar* species previously considered within the genus *Altingia.*

*Liquidambar* fossils from the Yongning Formation also show a combination of significant diagnostic features of three former genera of Altingiaceae. *Liquidambar nanningensis* is close to some extant species previously considered within the genus *Altingia* in having capitate infructescences composed of bilocular fruits lacking persistent styles and perianth, as well as in a small number of fruits per head not exceeding 10–12. However, the infructescences of *L. nanningensis* lack a cuplike bract at the base. In *L. nanningensis*, individual fruits are tightly pressed to each other, having no visible rim of tissues between fruits, as is also typical for the extant species *L. styraciflua* and *L. acalycina.* Leaves associated with *L. nanningensis* are most similar to the leaves of extant species previously considered within the genus *Semiliquidambar* in that they have a partly untoothed lamina base, an irregular distribution of teeth along the margin, and suprabasal venation. The pollen grains combine features of those of the former *Altingia* species (frequency and size of verrucae on the tectum, perforation size and shape in the reticulum, a smooth pore margin with rare minute verrucae, and relatively large verrucate granules of the pore membrane) and *Liquidambar* species (round and occasionally slightly elliptic pores, especially, in *L. formosana*). 

Fossil *Liquidambar* species were widely represented in the Eocene of South China in terms of richness and diversity, which suggests that the region may have been a center of *Liquidambar* speciation during this time [30,31,32]. The new *Liquidambar* fossils from the Nanning Basin expand our understanding of patterns of the genus diversification and possible ways of the migration of ancient sweetgums and can be regarded as paleobotanical justification for combining the three genera, *Liquidambar*, *Altingia*, and *Semiliquidambar*, into the single genus *Liquidambar* as proposed recently by Ickert-Bond and Wen [1].

## 4. Materials and Methods

*Liquidambar* fossils are found within the Nanning Basin at the locality named Erzhuanchang (22°53′10.44″ N, 108°26′0.22″ E; Figure 9A,B), Santang Town, Nanning City, Guangxi, South China. The Nanning Basin is an inland fault basin located in the southern part of Guangxi. The basement of this basin is composed of Cambrian, Devonian, and Carboniferous strata [70]. The basin infill consists of the Paleogene clastic deposits [71]. The lower part of the sedimentary succession (the Ducun and Yongjiang formations) is mainly composed of piedmont fluvial deposits, and the upper part (the Yongning Formation) consists of predominantly lacustrine coal-bearing deposits [70,72]. The Ducun and Yongjiang formations are only exposed along the edges of the basin while exposures of the Yongning Formation dominate and cover most of the basin [72]. The Yongning Formation is subdivided into the lower, middle, and upper parts based on their lithological composition. The lower part of the Yongning Formation has been dated to the early Oligocene on the basis of the occurrence of mammal fossils (*Heothema nanningensis* Zhao and *Heothema youngi* Zhao) [70,73]. The fossils described here were collected from the upper part of the Yongning Formation. The upper part of the formation is dated to the late Oligocene based on *Anthracotherium changlingensis* Zhao, *Anthracokeryx kwangsiensis* Qiu, and *Heothema* sp mammal fossil assemblages [72]. In recent years, a large number of plant fossils have been reported from the upper part of the Yongning Formation at the Santang fossil site, including three-dimensionally preserved fruits and wood [74,75,76,77,78]. 

Lithologically, the lower part of the outcrop at the Erzhuanchang locality is composed of bluish-gray mudstones with gastropod fossils, the middle part consists of plantbearing gray–black mudstones, and the upper layer is represented by yellow and purplish-red silty mudstones (Figure 9C,D).

The infructescences and leaves of *Liquidambar* were photographed using a Sony Alpha 6400 camera. For leaf description terminology, we follow the *Manual of Leaf Architecture* [79]. The infructescences were scanned with a Zeiss Xradia 520 Versa X-ray microscope at Nanjing Institute of Geology and Palaeontology, Chinese Academy of Sciences (Nanjing, China). Micro-computed tomography (micro-CT) image data were processed with VG Studio MAX software (v3.0). Dispersed pollen grains of *Liquidambar* were obtained from the rock sample containing the infructescence. Sediment samples were treated with 10% HCl, 30% HF, and 10% KOH, and then, the sample residues were sieved through a 5 μm nylon mesh in an ultrasonic cleaner. Pollen grains were examined using a Quanta 400 thermal field emission scanning electron microscope (SEM). Several *Liquidambar* pollen grains of the same type were found adhering to the surface of the carpels. The carpel outer cuticle and pollen grains on the cuticle were observed using an Olympus BX53 (LM) light microscope equipped with a UPlanFLN 100× digital camera. 

All fossil specimens are stored at the Museum of Biology, Sun Yat-sen University, Guangzhou, China.

## 5. Conclusions

The capitate infructescences, leaves, and pollen grains of *Liquidambar* from the Oligocene of the Nanning Basin in South China demonstrate a mosaic combination of morphological features embracing the former Althingiaceae genera *Liquidambar* s. str., *Altingia*, and *Semiliquidambar*. The infructescences of *L. nanningensis* are similar to those of extant species of the former genus *Altingia* in the number of fruits per head and the absence of styles. Fruiting heads of extant species of *Liquidambar* s. str. share with those of *L. nanningensis* the lack of a distinct peripheral rim between adjacent fruits and peculiarities in the fruit wall anatomy. Dispersed and attached to the carpels of *L. nanningensis* are pantoporate pollen grains that possess morphological features mainly characteristic of extant species of the former genus *Altingia*, although the varying pore shape is observed in pollen grains of extant species of *Liquidambar* s. str. Fossil leaves of *Liquidambar* from the Nanning Basin resemble those of the former genus *Semiliquidambar* and *Liquidambar* s. str. in shape, suprabasal venation, and the distribution of marginal teeth. Hence, these *Liquidambar* fossils can be regarded as paleobotanical evidence for the validity of combining the Altingiaceae former genera into one genus, *Liquidambar*. Based on an approach, known as the “whole plant concept” [80], we infer that the described plant fossils from the Nanning Basin might belong to the same plant. 

The first record of *Liquidambar* infructescences and leaves in the Oligocene of South China expands our understanding of the taxonomic diversity of extinct Altingiaceae and sheds new light on the paleobiogeographical and evolutionary history of the genus.

## Figures and Tables

**Figure 1 plants-13-00275-f001:**
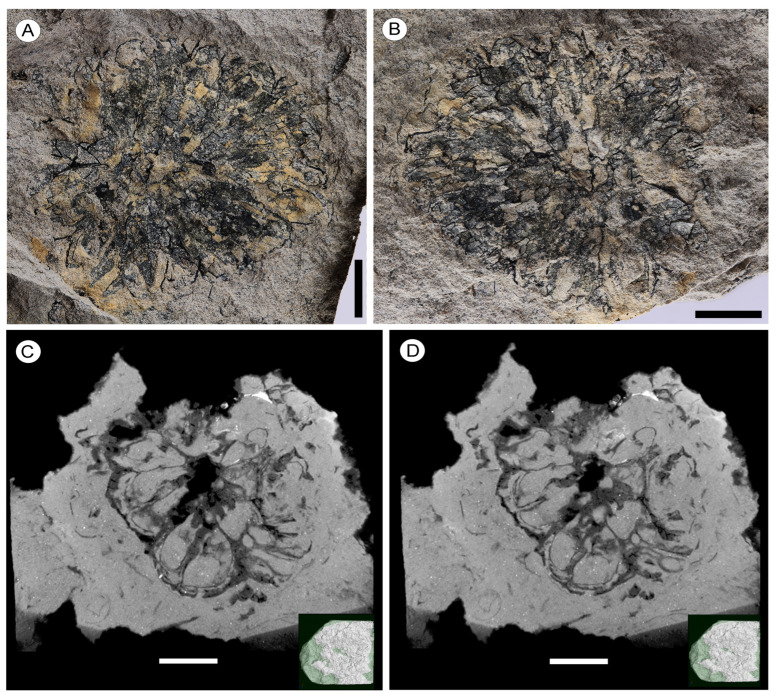
Capitate infructescences of *Liquidambar nanningensis*, holotype. (**A**,**B**) Subcircular infructescence; note the absence of persistent style bases, NNEZ-15a and NNEZ-15b, respectively. (**C**,**D**) Micro-CT images showing bilocular fruits, free distally and fused basally, NNEZ-15b. Scale bar: 5 mm.

**Figure 2 plants-13-00275-f002:**
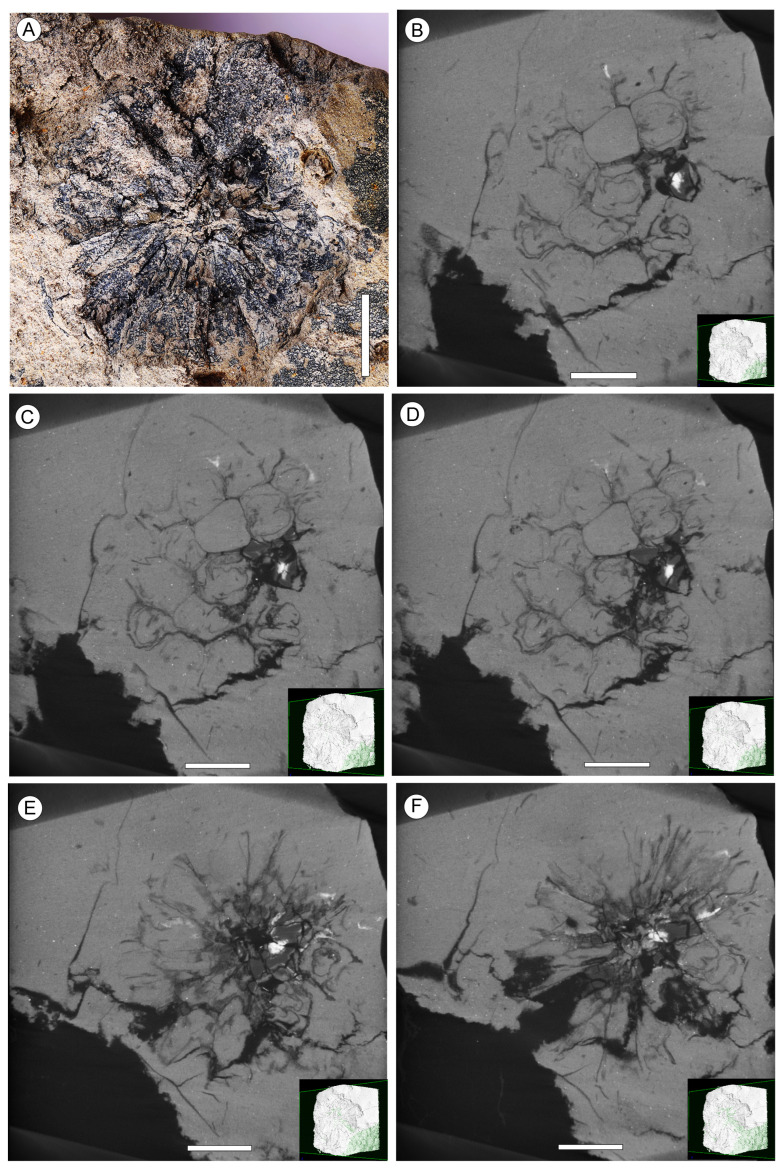
Capitate infructescences of *Liquidambar nanningensis*, NNEZ-21b. (**A**) Rounded infructescence. (**B**–**F**) A series of consecutive micro-CT images. Scale bar: 5 mm.

**Figure 3 plants-13-00275-f003:**
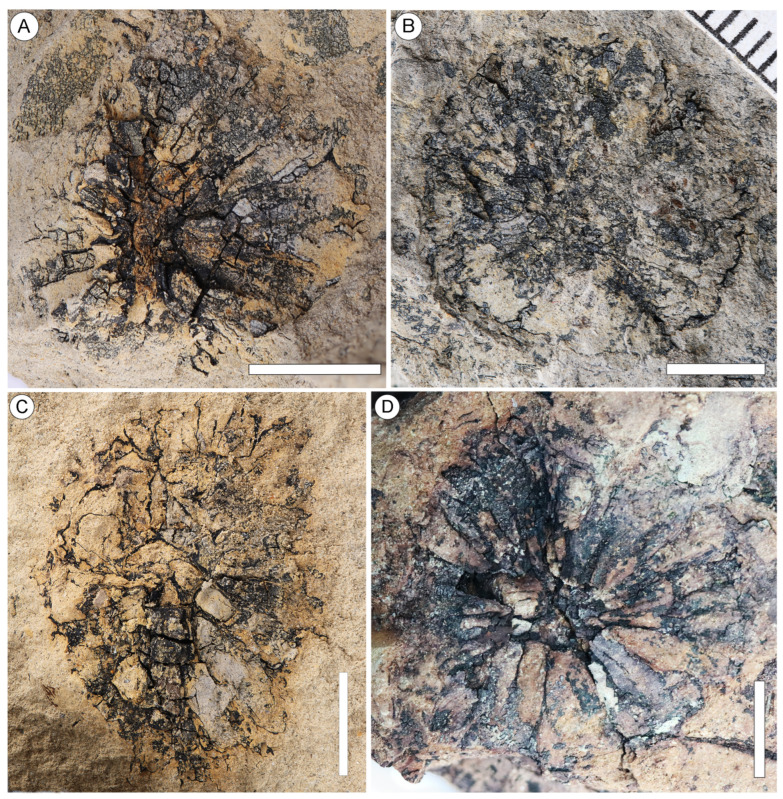
Capitate infructescences of *Liquidambar nanningensis*. (**A**) Helically arranged fruits, NNEZ-28. (**B**) Circular infructescence, NNEZ-32. (**C**) Rounded infructescence; note the tight connection between adjacent fruits, NNEZ-35. (**D**) The elongated obconical shape of carpels, NNEZ-42. Scale bar: 5 mm.

**Figure 4 plants-13-00275-f004:**
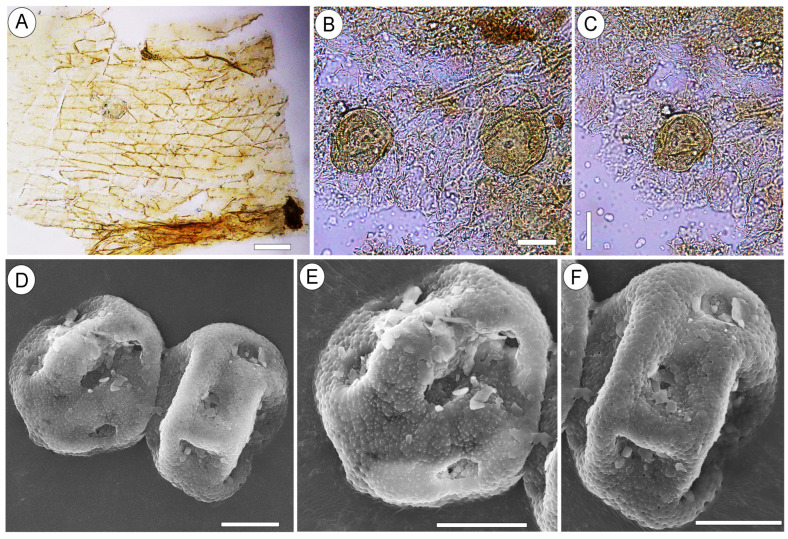
Epidermal characteristics of the *Liquidambar nanningensis* carpel wall and associated pollen grains of *Liquidambar*. (**A**) Carpel wall epidermal cells with oblique end walls, NNEZ-17a. (**B**,**C**) Pollen grains attached to the outer surface of the carpel wall cuticle, NNEZ-42. (**D**–**F**) Dispersed pantoporate pollen grains with circular and slightly elliptic pores obtained from the rock sample containing the infructescence, NNEZ-42. Scale bar: 50 µm (**A**), 20 µm (**B**,**C**), and 10 µm (**D**–**F**).

**Figure 5 plants-13-00275-f005:**
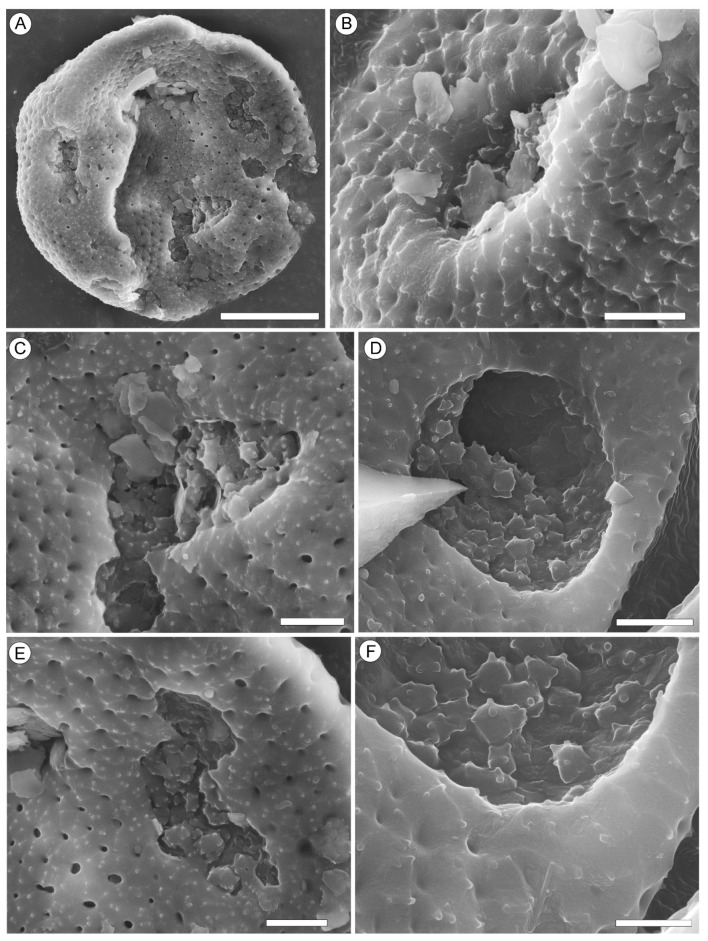
Morphology of the pollen grains of *Liquidambar* sp. 1, NNEZ-42, SEM. (**A**) Circular pantoporate pollen grain. (**B**–**D**) Round to slightly elliptic pores. (**E**,**F**) Details of the foveolate exine and pore membrane. Scale bar: 10 µm (**A**), 2 µm (**B**–**E**), and 1 µm (**F**).

**Figure 6 plants-13-00275-f006:**
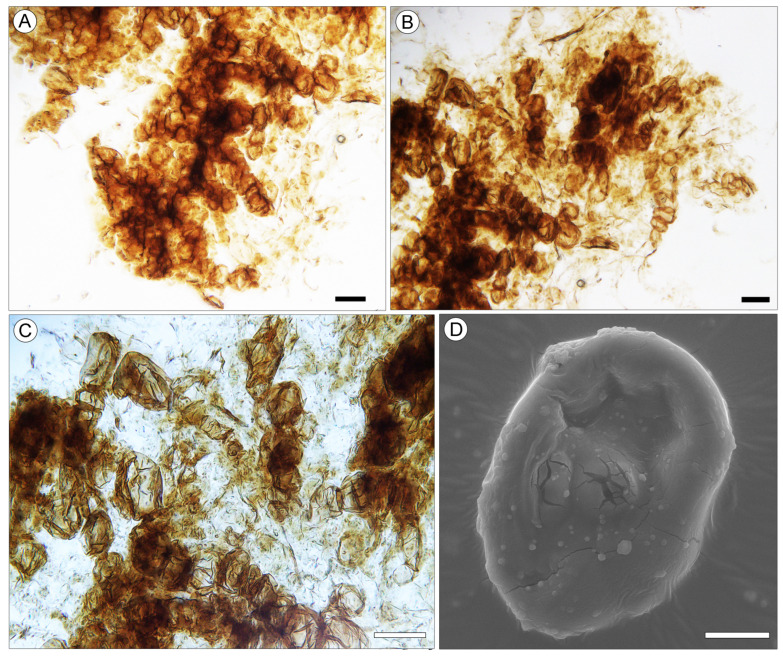
Epiphytic fungi on the *Liquidambar nanningensis* carpel surface. (**A**–**C**) Fungal spores, NNEZ-43, LM. (**D**) Fungal fruiting body, NNEZ-48, SEM. Scale bar: 20 µm (**A**–**C**) and 2 µm (**D**).

**Figure 7 plants-13-00275-f007:**
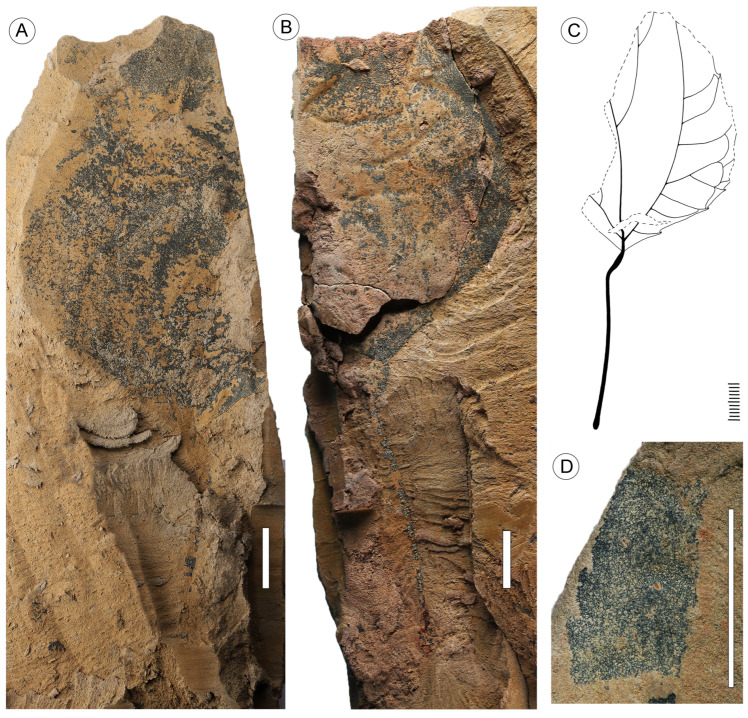
Leaf fossil of *Liquidambar* sp. 2 and line drawing of the leaf. (**A**) A leaf with a long petiole and three primary veins, NNEZ-45a-1. (**B**) A leaf with a serrate margin, NNEZ-45b, counterpart of NNEZ-45a-1. (**C**) Line drawing of a leaf, NNEZ-45b. (**D**) Toothed leaf margin, NNEZ-45a-2. Scale bar: 1 cm.

**Figure 8 plants-13-00275-f008:**
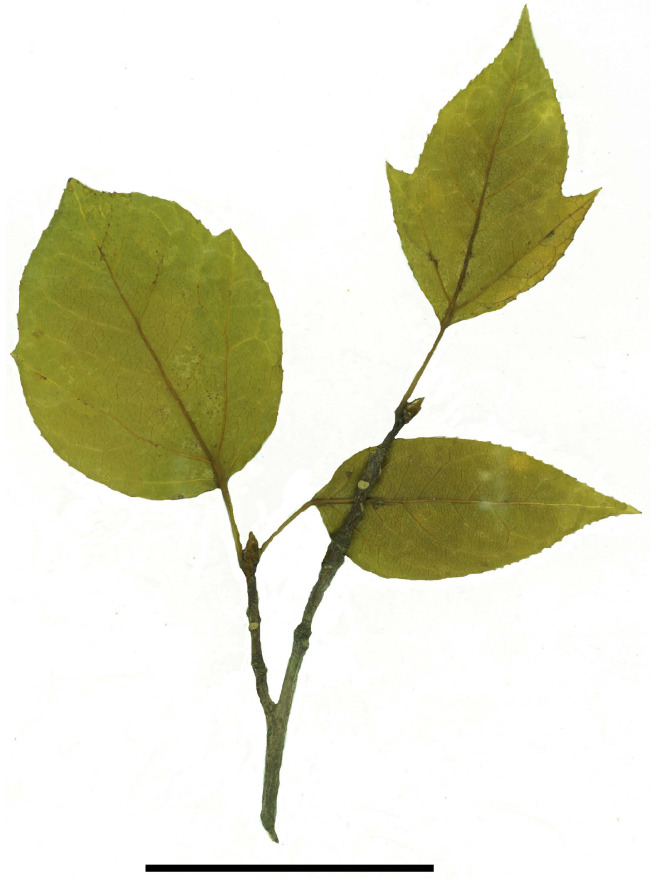
Shoot of the extant species *Liquidambar caudata*. Scale bar: 5 cm.

**Figure 9 plants-13-00275-f009:**
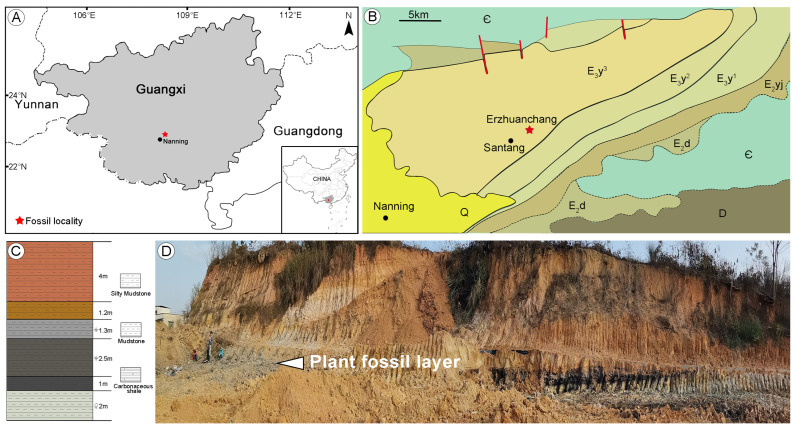
Location and stratigraphy of the fossil site in Guangxi, South China. (**A**) Geographic maps showing the location of the Nanning Basin. (**B**) Geological map of the Nanning Basin, modified from Zhao (1981) and Quan et al. (2016) [70,74]. Ꞓ—Cambrian, D—Devonian, E_2_d—Ducun Formation, E_2_yj—Yongjiang Formation, E_3_y^1^—lower part of the Yongning Formation, E_3_y^2^—middle part of the Yongning Formation, E_3_y^3^—upper part of the Yongning Formation, and Q—Quaternary. (**C**) Lithological column of the fossil site. (**D**) Outcrop of the Erzhuanchang locality. The arrow indicates the plant fossil layer.

## Data Availability

Data are contained within the article.

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
