# Peer review of "Structurally Preserved Liquidambar Infructescences, Associated Pollen, and Leaves from the Late Oligocene of the Nanning Basin, South China"

_plants, 2024, doi:10.3390/plants13020275_

Round 1

Reviewer 1 Report

Comments and Suggestions for Authors

The biological aspects of the work are notably fascinating from a biological standpoint. However, a significant issue arises from the absence of a proper geological framework in the presentation of the fossil record. It is imperative to incorporate a comprehensive geological map that clearly indicates the site's location. Additionally, the introduction should provide justification for the procedures employed to date these strata as Oligocene. Failure to address this concern will result in a decontextualization of crucial data, rendering it invalid. Therefore, it is essential to rectify this issue to ensure the overall integrity and reliability of the study.

Author Response

Review 1

The biological aspects of the work are notably fascinating from a biological standpoint. However, a significant issue arises from the absence of a proper geological framework in the presentation of the fossil record. It is imperative to incorporate a comprehensive geological map that clearly indicates the site's location. Additionally, the introduction should provide justification for the procedures employed to date these strata as Oligocene. Failure to address this concern will result in a decontextualization of crucial data, rendering it invalid. Therefore, it is essential to rectify this issue to ensure the overall integrity and reliability of the study.

Replay: The Material and Methods section of the manuscript has been revised to include information about the geographical location and stratigraphic age of the fossils found. At the same time, a geological map of Nanning Basin has been added to Figure 1 to show the location of the fossil site.

Reviewer 2 Report

Comments and Suggestions for Authors

The article is well written I have the following minor suggestions on this article.
1. Consider adding more specific details about the morphological and anatomical characteristics used to describe the new species, Liquidambar nanningensis sp. nov.

2. Provide a brief explanation or list of the features that indicate the relationship between Liquidambar and the former genera of Altingiaceae, Altingia, Liquidambar s. str., and Semiliquidambar.

3. Discuss the significance of the new occurrence and its implications for the taxonomic and morphological diversity of Paleogene Liquidambar species in South China. Highlight any insights into the evolution or distribution patterns of Liquidambar.

Comments on the Quality of English Language

English is fine.

Author Response

Review 2

The article is well written I have the following minor suggestions on this article.

1. Consider adding more specific details about the morphological and anatomical characteristics used to describe the new species, Liquidambar nanningensis sp. nov.

Replay: We have tried to study all the currently available morphological and anatomical features of these fossil capitate reproductive structures using different methods of study.

2. Provide a brief explanation or list of the features that indicate the relationship between Liquidambarand the former genera of Altingiaceae, Altingia, Liquidambars. str., and SemiLiquidambar.

Reply: We have added a brief description of the morphological features of the former genera of Altingiaceae and moved this part from the Discussion section to the beginning of Introduction.

3. Discuss the significance of the new occurrence and its implications for the taxonomic and morphological diversity of Paleogene Liquidambarspecies in South China. Highlight any insights into the evolution or distribution patterns of Liquidambar.

Replay: We have added a paragraph at the end of section ‘Taxonomic significance of new Liquidambar fossils from the Nanning Basin’ with a brief discussion on the genus diversification and taxonomic significance. Significance of Liquidambar species from South China for plant taxonomy, phylogeny and phytogeography is discussed in more details in Maslova et al., 2023.

Reference: Maslova NP, Kodrul TM, Kachkina VV, et al. Evidence for the evolutionary history and diversity of fossil sweetgums: leaves and associated capitate reproductive structures of Liquidambar from the Eocene of Hainan Island, South China. Papers in Palaeontology, 2023, 9(6): e1540.

Reviewer 3 Report

Comments and Suggestions for Authors

This manuscript reported a new species of Liquidambar from the late Oligocene of the Nanning Basin, South China. It is based on some well-preserved infructescences, associated pollen and leaves. The materials are carefully described and compared with fossil and extant Asian Liquidambar species. This new discovery of Liquidambar expands the taxonomic and morphological diversity of the Paleogene Liquidambar species in South China. The manuscript is well prepared with accurate identification, description on the infructescences. Therefore, my recommendation for this manuscript is minor revision.

Comments on the Quality of English Language

The abstract, comparisons, discussions and figures need some minor improvements. The comments are as follows:

1.In line 16–18, The sentence “This study presents the first occurrence of in fructescences, associated pollen and leaves of Liquidambar in the upper Oligocene of the Nanning Basin in South China.” is unclear. Is this the first occurrence of the combination of different organs of the genus in the whole fossil record? Or, is it in any certain age or region? Therefore, rewriting of this sentence is suggested.

2. In line 86–90, the holotype is mentioned earlier than the other specimens, so “Figure 3A-D” is suggested to change into “Figure 2A-D” and the “Figure 2” in the text change into “Figure 3”.

3. The pollen grain in Figure 5C looks like Alnus pollen with 5 pores and very strong arch between the pores. Please, if possible, check under the SEM to make sure if it is Alnus pollen.

4. In line 123, remove the sentence “Two fragmentary preserved leaves of Liquidambar are associated with infructescences.”.

5. In figure 8, we can clearly see the intumescent petiole near the base of the leaf. So, it is doubtful to assign it to Liquidambar leaf. Please check it carefully.

Author Response

Review 3

The abstract, comparisons, discussions and figures need some minor improvements. The comments are as follows:

1. In line 16–18, The sentence “This study presents the first occurrence of in fructescences, associated pollen and leaves of Liquidambar in the upper Oligocene of the Nanning Basin in South China.” is unclear. Is this the first occurrence of the combination of different organs of the genus in the whole fossil record? Or, is it in any certain age or region? Therefore, rewriting of this sentence is suggested.

Replay: Thank you for your suggestions. This sentence has been rewritten to ‘This study presents the first occurrence of Liquidambar in the Oligocene of South China. Fossil sweetgum infructescences, associated pollen and leaves have been found in the Nanning Basin, Guangxi.’

2. In line 86–90, the holotype is mentioned earlier than the other specimens, so “Figure 3A-D” is suggested to change into “Figure 2A-D” and the “Figure 2” in the text change into “Figure 3”.

Replay: The numbers of figures have been changed orderly

3. The pollen grain in Figure 5C looks like Alnus pollen with 5 pores and very strong arch between the pores. Please, if possible, check under the SEM to make sure if it is Alnus pollen.

Replay: We have adjusted the Figure to remove the picture of this grain of pollen. We have no further information about this pollen grain at this time and whether it is assigned to Alnus remains to be confirmed.

4. In line 123, remove the sentence “Two fragmentary preserved leaves of Liquidambarare associated with infructescences.”

Replay: This sentence has been removed here.

5. In figure 8, we can clearly see the intumescent petiole near the base of the leaf. So, it is doubtful to assign it to Liquidambar leaf. Please check it carefully.

Replay: The thickening of the petiole may be due to insect injury. Based on the existing morphological characteristics of leaf shape and vein, it can be classified as Liquidambar.

Round 2

Reviewer 1 Report

Comments and Suggestions for Authors

No comment